# Dynamic Residual Dense Network for Image Denoising

**DOI:** 10.3390/s19173809

**Published:** 2019-09-03

**Authors:** Yuda Song, Yunfang Zhu, Xin Du

**Affiliations:** 1Information Science & Electronic Engineering, Zhejiang University, Hangzhou 310027, China; 2Computer Science & Information Engineering, Zhejiang Gongshang University, Hangzhou 310027, China

**Keywords:** noise reduction, image restoration, deep learning, dynamic network

## Abstract

Deep convolutional neural networks have achieved great performance on various image restoration tasks. Specifically, the residual dense network (RDN) has achieved great results on image noise reduction by cascading multiple residual dense blocks (RDBs) to make full use of the hierarchical feature. However, the RDN only performs well in denoising on a single noise level, and the computational cost of the RDN increases significantly with the increase in the number of RDBs, and this only slightly improves the effect of denoising. To overcome this, we propose the dynamic residual dense network (DRDN), a dynamic network that can selectively skip some RDBs based on the noise amount of the input image. Moreover, the DRDN allows modifying the denoising strength to manually get the best outputs, which can make the network more effective for real-world denoising. Our proposed DRDN can perform better than the RDN and reduces the computational cost by 40–50%. Furthermore, we surpass the state-of-the-art CBDNet by 1.34 dB on the real-world noise benchmark.

## 1. Introduction

Deep convolutional neural networks have demonstrated their effectiveness on various image restoration tasks (e.g., denoising and super-resolution). Specifically, the RDN obtains state-of-the-art performance on image noise reduction with multiple noise levels. We rethink the reason why the RDN achieves success on various image restoration tasks. The RDB is a combination of the residual block [1] and dense block [2]. It has been proved that the dense block can improve image processing by making full use of multi-scale information [3,4,5]. Meanwhile, the residual connection can make deep neural networks easier to train and reuse features in forward propagation [6]. Besides, the RDN adaptively fuses hierarchical features from all RDBs, which is called global feature fusion. We believe that residual connection and global feature fusion will benefit more to RDN.

**Residual Connection.** In addition to the above properties, residual networks can also be viewed as a collection of many paths that do not strongly depend on each other [7]. Therefore, some researchers discard the residual blocks that contribute little to the results [8]. Considering the similarity between the RDB and residual block, we believe that some RDBs can also be discarded, even if the RDB contains more convolutional layers.

**Deep Supervision.** Deep supervision [9] is widely used in various computer vision tasks (e.g., classification, action recognition). Especially, Lee et al., proposed UNet++ [10], which uses deep supervision to achieve better results. UNet++ can be manually pruned based on the difficulty of segmentation tasks to reduce computational cost while inferring via deep supervision. Although not the same as UNet++, the RDN also uses deep supervision by a concatenation followed by point-wise convolution.

Inspired by the above properties, we presume that some RDBs in the RDN help little while inferring. We realize that the outputs of some of the adjacent RDBs are similar via visualizing the output of each RDB in the RDN. This means that the RDBs are redundant. Figure 1 shows the output feature maps of 1-st to 20-th RDB in RDN and the 0-th feature map denotes the input feature map of 1-st RDB. Specifically, we regard the mean of all channels of the feature maps as the approximation of the feature maps to achieve the visualization. As shown in Figure 1, the feature maps of some adjacent RDBs are similar (similar feature maps are in the same red box), and some can be considered to differ only in mean and variance. Similar feature maps mean that these RDBs contribute little to the entire network. We believe that replacing these RDBs with identity mappings can still yield comparable processing results same as the residual blocks shown in [8]. Moreover, this allows the RDN to prune the network to reduce computational cost based on the difficulty of denoising like UNet++.

UNet++ prunes the network based on the degree of deterioration of the results on the test dataset, but we want the network to automatically determine the difficulty of denoising and select the blocks that need to be pruned [11]. Furthermore, we observe that the number of pruned blocks in the RDN will affect the strength of denoising. In other words, pruning the partial blocks will not only reduce the computational cost but also fine-tune the strength of the denoising. This is meaningful because a slightly weaker and slightly stronger denoising strength is needed to get the results that most satisfy human perception.

The RDN cascades dozens of RDBs, and each RDB extracts local features via dense connected convolutional layers, which consume a lot of memory and computational cost while inferring. Therefore, we treat the cascaded blocks as a sequence and use a gate module containing the RNN to predict the importance of the next block for processing the current feature map. If the importance of the prediction is below the threshold, we skip the next block and degenerate the block into an identity function. Furthermore, the RDN only performs great in denoising on a single noise level since the RDN does not have a module for estimating real-world noise levels. The gate module frees the RDN from the noise level estimation. Moreover, it makes the RDN achieve an adjustable level of denoising strength to better satisfy human perception. Our proposed network can dynamically adjust the number of blocks used in the RDN based on the noise amount, and therefore we call it the *dynamic residual dense network*.

## 2. Related Work

### 2.1. Denoising

Image denoising is one of the most extensively studied topics in image processing. There are many types of image denoising methods, such as filtering i.e., BM3D [12]), effective prior i.e., EPLL [13]) and low-rank i.e., WNNM [14]). Recently, deep convolutional neural networks have improved denoising performance. Zhang et al., [15] proposed the DnCNN to embrace the process in a very deep architecture. Specifically, global residual learning is used for boosting the denoising performance. A novel residual dense network (RDN) was proposed by Zhang et al., [3] to make full use of hierarchical features from the original low-quality images. The residual dense block (RDB) is used for extracting abundant local features. Some methods combine the traditional pipeline to obtain better performance. NLCNN [16] and UDNet [17] combine the deep neural network and the block matching. This makes the networks obtain both local and global information. Inspired by k-nearest neighbors (KNN), Tobias et al., [18] proposed the neural nearest neighbors block ($N^{3}$ block), which leverages the principle of self-similarity and can be viewed as an embedding block in network architectures. Motivated by the persistency of human thoughts, Tai et al., [19] proposed a memory network that introduced a memory block to explicitly mine persistent memory through an adaptive learning process. Liu et al., [20] presented a U-Net like multi-level wavelet CNN (MWCNN) to balance the receptive field size and computational efficiency; their method attained good performance on the image denoising task. The aforementioned image denoising methods have achieved promising results, but current CNN-based denoising technologies suffer from specific drawbacks. For example, they are not effective for real-world noisy images. To overcome this, Guo et al., [21] proposed a realistic noise model and a network called CBDNet for blind denoising of real-world noisy images. CBDNet obtained state-of-the-art results on multiple benchmarks of real-world noisy images. Moreover, CBDNet significantly outperformed the non-blind denoisers on denoising of real-world noisy images. Chen et al., [22] proposed a novel framework called DBF which combines deep learning technique with boosting algorithm. The DBF uses the CNN to build the boosting units which make the DBF obtain impressive performance and speed on real-world noise reduction. Priyanka et al., [23] proposed a novel architecture with a chain of successive symmetric convolutional-deconvolutional layers called FSCN. It applies a deep fully symmetric convolutional-deconvolutional neural network and an adaptive moment optimizer to achieve superior denoising.

### 2.2. Datasets

Real-world noise datasets or benchmarks are important to real image noise reduction. Researchers generally add the additive white Gaussian noise (AWGN) to clear images to obtain corresponding noisy images. Then the image pairs can be used to evaluate the performance of the denoising methods or train a CNN-based denoising model. BSD68 [24], BSDS [25] and Set5/Set14 [26] are all widely used image datasets. Although the AWGN is an excellent noise model, it is still somewhat different from real-world noise. RENOIR [27] is a dataset for real low-light image noise reduction. It consists of images of 120 scenes taken by three cameras. Each scene contains 1 ground truth image and 1 or 2 noisy images. All images in RENOIR were taken in low-light conditions. Ground truth images were taken with low ISO and long exposure times, and noisy images were taken with high ISO and short exposure times. Abdelhamed et al., [28] released a high-quality denoising dataset (namely, SIDD). Different from RENOIR, the images in SIDD were taken by five smartphone cameras. Furthermore, SIDD contains images taken in the same scene with different lighting conditions. It contains multiple image pairs in each scene instance for better denoising. The Darmstadt Noise Dataset (DND) is a novel benchmark for evaluating methods used in real-world noise reduction. Different from RENOIR and SSID, DND does not provide ground truth images. Moreover, users can only upload the processing results to the online evaluation website. It is a fair benchmark for evaluating the performance of denoising methods. Besides, the Nam dataset [29] and the PolyU dataset [30] are also great benchmarks for real-world noisy images denoising. The Nam dataset contains 500 image pairs in 11 scenes and the PolyU dataset contains 2008 image pairs in 40 scenes. Different from the DND, the Nam dataset and the PolyU dataset store the images in Joint Photographic Experts Group (JPEG) format.

## 3. Methodology

We first present the overall architecture of our proposed DRDN in Section 3.1. Then we will highlight the mechanism for dynamically skipping some RDBs in Section 3.2. Finally, we provide the training details in Section 3.3.

### 3.1. Architecture of DRDN

As shown in the top part of Figure 2, our proposed DRDN mainly consists of four parts: a feature extraction module, dynamic residual dense blocks (DRDBs), feature fusion, and a reconstruction module. Specifically, the feature extraction module uses two convolutional layers to extract shallow  features:(1)$$\begin{array}{cc} F_{-1} & =H_{FE1}\left(I_{LQ}\right),\\ F_{0} & =H_{FE2}\left(F_{-1}\right), \end{array}$$
where $H_{FE1}$ and $H_{FE2}$ are the convolutional layers, and $I_{LQ}$ refers the noisy image.

Different from the RDN, the DRDN uses DRDBs to extract local features, which can be obtained by the following:(2)$$\begin{array}{cc} F_{d} & =B_{d}\left(F_{d-1}\right)\\ \; & =B_{d}\left(B_{d-1}\left(\cdots \left(B_{1}\left(F_{0}\right)\right)\cdots \right)\right), \end{array}$$
where $B_{d}$ denotes the d-th DRDB. We will detail the structure of the DRDB later.

We then use feature fusion to make full use of multi-scale information after extracting local features. As previously mentioned, feature fusion is also a deep supervision module. The global feature fusion can be obtained by
(3)$$F_{DS}=H_{DS}\left(\left[F_{1},\cdots ,F_{D}\right]\right),$$
where [⋯] denotes the concatenation operation, and $H_{DS}$ is a point-wise convolutional layer.

Like the feature extraction module, the reconstruction module also uses two convolutional layers to reconstruct the image and uses global residual learning to obtain high-frequency features more easily:(4)$$\begin{array}{cc} F_{DF} & =H_{GFF}\left(F_{DS}\right)+F_{-1},\\ I_{HQ} & =H_{FR}\left(F_{DF}\right)+I_{LQ}, \end{array}$$
where $H_{GFF}$ is a convolutional layer that can further fuse multi-scale local features, $H_{FR}$ is a convolutional layer used to reconstruct the image, and $I_{HQ}$ refers the denoised image.

### 3.2. Dynamic Residual Dense Block

The bottom part of Figure 2 presents the details of our proposed dynamic residual dense block (DRDB). Each DRDB contains two paths and uses the gate module to make a binary decision to determine which path is selected. One of the paths is an identity function, and the other consists of three parts: a dense block, local feature fusion, and local residual learning. This is the same as in the RDB.

The dense block in the DRDB contains *C* convolutional layers, and the c-th convolutional layer can be obtained by the following:(5)$$F_{d,c}=\sigma \left(H_{d,c}\left[F_{d-1},F_{d,1},\cdots ,F_{d,c-1}\right]\right),$$
where σ represents the ReLU activation function [31], and $F_{d,c}$ denotes the c-th convolutional layer in the d-th DRDB. Local feature fusion is similar to global feature fusion, and the output of the d-th DRDB can be formulated as follows: (6)$$F_{d,LF}=H_{d,LFF}\left(\left[F_{d-1},F_{d,1},\cdots ,F_{d,C}\right]\right).$$

Local residual learning is similar to global residual learning, and can be obtained by
(7)$$F_{d}^{0}=F_{d,LF}+F_{d-1}.$$

As shown in Figure 2, the gate module contains a convolutional layer to extract features for the decision, followed by a global average pooling [32] and a point-wise convolutional layer to reduce the dimension of the tensor. The convolutional layer, the global average pooling and the point-wise convolutional layer process the previous feature map, retaining only the information useful for the binary decision for whether to skip the followed block. The information retained may include noise levels and other important characteristics. The procedure can be considered as a bottleneck layer and it can be obtained by
(8)$$D_{d}=H_{P}\left(P_{GA}\left(H_{C}\left(F_{d-1}\right)\right)\right),$$
where $D_{d}$ denotes d-th the output of the bottleneck layer, and $H_{C}$, $P_{GA}$ and $H_{P}$ represent the convolutional layer, the global average pooling and the point-wise convolutional layer. Then, we use an LSTM [33] to make the decision information flow between blocks. Because the gate modules are actually to solve a sequential decision task, the LSTMs play a important role in the decision. LSTM allows the current gate module to remember the choice of the previous gate modules. When the previous gate modules choose to skip the followed blocks, the current gate module will appropriately reduce the skip probability. Finally, we apply a fully connected layer and a sigmoid function followed by a binary step function to make the binary decision.

Since the gradient cannot effectively flow in the binary step function, we only apply the binary step function in forward propagation and skip it in backward propagation. The procedure can be obtained by
(9)$$G\left(x\right)=\left\{\begin{array}{c} \delta (S(x)-t)\\ S(x) \end{array}\begin{array}{c} ,\;\mathrm{forward}\\ ,\;\mathrm{backward} \end{array}\right.,$$
where *t* is the threshold, and δ(x) and S(x) represent the binary step function and the sigmoid function, respectively.

Finally, the output of each DRDB is determined by the gate module. When the output of the gate module is 0, the output of the d-th DRDB is the same as the output of the {d−1}-th DRDB. When the output of the gate module is 1, we take the output of the residual dense block as output. More precisely, this can be formulated as
(10)$$F_{d}=\left\{\begin{array}{c} F_{d-1}\\ F_{d}^{0} \end{array}\right.\begin{array}{c} ,\;G_{d}\left(v_{d}\right)=0\\ ,\;\mathrm{otherwise} \end{array},$$
where $v_{d}$ denotes the output vector of the fully connected layer in the d-th gate module.

When training, the gradient propagation in the DRDB is affected by the decision of the gate modules. When the output of the gate module is 0, the gate module degenerates the followed block into an identity function. This means that the gradient of the followed block will not be computed. When the output of the gate module is 1, the followed block will further process the previous feature map, and its gradient will be computed. No matter what the decision of the gate module is, the gradient in the gate module will always propagate.

When inferring, since the network only performs forward propagation, there is no need to get the output of the residual dense block if the output of the gate module is 0. We can collect the outputs of all gate modules to count the number of the residual blocks skipped. We calculate the ratio of the blocks that are skipped to the total blocks, which is called the skip ratio. Meanwhile, the skip ratio and the strength of the denoising of the network can be changed by modifying the threshold *t*. However, since the threshold and the skip ratio are not consistent, we use a simple linear transformation to map the threshold to the skip ratio:(11)t=ωλ+(1−ω)ε,
where λ denotes the skip ratio, and ω and ε are the parameters adjusted based on the trained model. We use fixed parameters (ω=0.1,ε=0.49) for generalization.

### 3.3. Training Algorithm

The training process is composed of three stages. We use the same loss function in the each stage, but the parameters of each stage are different. The loss function consists of the $L_{1}$ loss and the average output probability of the gate modules. The loss function can be formulated as follows:(12)$$L={∥y-\hat{y}∥}_{1}+\alpha \frac{1}{N}\sum_{d=1}^{N}S_{d}\left(v_{d}\right),$$
where *y* and $\hat{y}$ denote the output image and ground truth image, and α is used to balance the effect and the computational cost.

**Stage 1.** Because the gradient propagation in the DRDB is affected by the decision of the gate modules, a simple pre-training is necessary to ensure that each DRDB obtains the ability of denoising. We fix the output of all gate modules to 1 i.e., the DRDN does not skip any dense residual block). Next, we use the orthogonal matrix [34] to initialize the parameters of the convolutional layers. For this stage, since we are training an RDN approximately and do not need to consider the gate modules, we set 0 as α in training. Because the gate modules are not involved in sequential decisions, their gradient won’t be computed.

**Stage 2.** We also set 0 as α in training, but no longer fix the output of the gate modules. We initialize the parameters of gate modules to small values to make the output close to 0 at the beginning. After the training of Stage 1, the DRDBs have gained the basic ability of denoising. And after this stage (including this stage), the gate modules participate in gradient propagation. This makes the gate modules learn the ability of extracting useful information. Finally, the DRDN will get the appropriate parameters for the binary decision.

**Stage 3.** We set α to $10^{-4}$ to guide the gate modules to skip more unimportant blocks. After training of Stage 1 and Stage 2, all blocks learn the ability to denoise, and depend little on each other. However, we do not effectively guide the decisions of the gate modules. Therefore, we activate the regularization term of the loss function and directly constrain the output of the gate modules. For this stage, the learning rate needs to be made smaller after the skip ratio is stabilized. This can make the DRDN get better denoising ability.

**Optional.** We also use reinforcement learning as the 4-th stage to train the gate modules to further improve the skip ratio. However, in our experiments, reinforcement learning may slightly deteriorate the results and limit the ability of the network to change the denoising strength by adjusting the threshold. In contrast, the DRDN can obtain a higher skip ratio by reinforcement learning. Therefore, we regard this stage as an optional training stage. Specifically, the reward of the d-th DRDB can be formulated as follows:(13)$$r_{d}=-\left[{∥y-\hat{y}∥}_{1}-\beta \left(1-G_{d}\left(v_{d}\right)\right)\right],$$
where β is used to adjust the influence of the supervised loss. We set $10^{-4}$ as β in the experiment. We apply the REINFORCE algorithm [35] in this stage. The algorithm is shown in Algorithm 1. The $s_{1}^{\left(k\right)}$, $a_{1}^{\left(k\right)}$ and $r_{1}^{\left(k\right)}$ denote the stage, the action and the reward for k-th image pairs. And γ is the learning rate.

**Algorithm 1** REINFORCE
Get a batch of *K* image pairs, $\left\{x^{\left(1\right)},y^{\left(1\right)}\right\},\cdots ,\left\{x^{\left(K\right)},y^{\left(K\right)}\right\}$With policy π, derive $\left(s_{1}^{\left(k\right)},a_{1}^{\left(k\right)},r_{1}^{\left(k\right)},\cdots ,s_{N}^{\left(k\right)},a_{N}^{\left(k\right)},r_{N}^{\left(k\right)}\right)$Compute gradients using REINFORCE
$$\Delta \theta =\frac{1}{K}\sum_{k=1}^{K}\sum_{i=1}^{N}\nabla_{\theta }\log\pi \left(a_{i}^{\left(k\right)}|s_{i}^{\left(k\right)};\theta \right)\left(\sum_{j=i}^{N}r_{j}^{\left(k\right)}\right)$$
Update parameters θ←θ+γΔθ


## 4. Experiments

### 4.1. Setting

In our proposed DRDN, we set 20 as the DRDB number, and 6 as the number of convolutional layers per DRDB. All convolutional layers and point-wise convolutional layers have 64 filters; however, in the DRDB, the convolutional layers have 32 filters (i.e., we set 32 as the growth rate). We use ADAM [36] with $\beta_{1}=0.9$ and $\beta_{2}=0.999$ to train our DRDN. We set the learning rate to $10^{-4}$ and train 200 epochs for stage 1 and 800 epochs for stage 2. In stage 3, we also start training at a learning rate of $10^{-4}$ but modify the learning rate to $10^{-5}$ after the skip ratio is stabilized. In general, we train DRDN for 1000 epochs in stage 3.

Instead of using a large batch size but small patch-size images as input, we prefer to use large patch-size images to train the network since this can make full use of the large receptive field of the DRDN. In each iteration, we randomly extract a 192×192 patch from the noisy image and the ground truth image simultaneously. We extract a patch pair from each image pair in each epoch, and thus different datasets have different iterations for each epoch. For better generalization, we apply various data augmentations to process the patches, including flipping, rotating, and cropping.

### 4.2. Local Performance Evaluation

In this section, we directly apply denoising on real-world noisy images because we believe that the non-blind denoising of Gaussian noise does not represent the performance of the denoising methods. We use the RENOIR and the SIDD for training and testing. We use the middle dataset of the SIDD, which contains two pairs of images in each scene instance. We compare the improved RDN, called RDN+ [37], with the DRDN in this section. For fairness, RDN+ also uses the same settings as the DRDN. We split the datasets into training sets and testing sets based on the cameras, and we only use sRGB images in the experiment.

We compare the effectiveness and efficiency of RDN+ and DRDN. We use the PSNR and SSIM to quantitatively analyze the denoising ability of these two denoising methods. FLOPs are applied to show the efficiency of the methods. The FLOPs of the DRDN is represented by the average FLOPs calculated from the skip ratio. In addition to the residual blocks, the average FLOPs of all modules in the DRDN are constant. The structure and the shape of the feature map of residual blocks are same. Therefore, we use the average skip ratio and the FLOPs of residual blocks to obtain the average FLOPs of the residual blocks. Finally, we add the average FLOPs of residual blocks and the FLOPs of the other modules to obtain the final average FLOPs. We also calculate the total number of parameters.

The testing results are shown in Table 1. The gate module only accounts for less than 3% of the total parameters in the DRDN. However, the DRDN achieves comparable performance to RDN+ on the two datasets and reduces the computational cost by 40–50%. Furthermore, the DRDN performs better than RDN+ and achieves lower FLOPs when test on SSID. This may be because the SIDD contains images taken in the same scene with different lighting conditions, and thus the noise amount of the images in SSID vary largely. This illustrates that the DRDN can better denoise real-world noisy images with different noise amounts. Finally, we evaluated the inference latency of the RDN+ and DRDN on an Nvidia Geforce RTX 2070 GPU. Because the DRDN only adds the gate modules before every residual blocks, the DRDN maintains the computational parallelism of residual blocks. This makes the DRDN to reduce the inference latency rather than just reducing the FLOPs.

Furthermore, we visualize the skip ratio of different blocks when testing. The skip ratio of each block test on the two datasets are shown in Figure 3. Overall, the later the block, the higher its average skip ratio. The apparently uneven skip ratio is caused by deep supervision. Deep supervision will fuse the outputs of every blocks, and the outputs of the former blocks will affect the later blocks. When the former blocks are skipped, the effect of skipping is added to all subsequent blocks. In other words, the former blocks play a more important role than the latter blocks in the DRDN. Therefore, the DRDN prefers to prune the later blocks.

### 4.3. Evaluation on the Real-world Noise Dataset

We obtain evaluation results on the sRGB track of the Darmstadt Noise Dataset [38]. For better generalization, we fine-tune the DRDN trained on the RENOIR by training it for 200 epochs on a mixed dataset. The mixed dataset contains 120 pairs of images from the RENOIR and 160 pairs of images from the SIDD. We conduct both quantitative and qualitative comparisons for denoising on DND.

We compare the methods of other published works on the DND benchmark website. The PSNR/SSIM results are listed in Table 2. Since DND provides the noise *std.* for better denoising, we further divide the methods into blind and non-blind. Obviously, traditional denoising methods, such as BM3D [12], perform poorly on real-world denoising. Some CNN-based methods have dramatically better denoising performance than traditional methods, such as DnCNN [15] and FFDNet [39]. Specifically, CBDNet achieves state-of-the-art performance on real-world denoising via asymmetric learning and a realistic noise model. Recently, some blind real-world denoising methods have been proposed and they perform better than CBDNet [40,41]. Benefitting from the automatic selection of denoising strength, our proposed DRDN achieves much better performance than CBDNet. Compared with Path-Restore [41], DRDN achieves a high PSNR but slightly lower SSIM.

Figure 4 further shows the qualitative results. BM3D fails to remove the noise in images. DnCNN+ removes the noise, but the texture in images is not well preserved. CBDNet performs well, but some edges are blurry. Path-Restore and DRDN perform equally well, but DRDN performs better on the processing of boundaries while Path-Restore retains more details. These findings are consistent with the quantitative results.

We apply the Nam dataset [29] and the PolyU dataset [30] for further evaluation. The images in the Nam dataset and the PolyU dataset are saved in JPEG format, which introduces lossy compression. Therefore, we converted the images in the mixed dataset to JPEG format to obtain a new mixed dataset and fine-tune the DRDN on the new mixed dataset for 100 epochs. The PSNR/SSIM results are listed in Table 3. Different from the results on the DND dataset, the DnCNN performs poorly on these two datasets. The reason for the significant degradation of the DnCNN is that the DnCNN is trained on the lossless compressed images, which makes the DnCNN not robust to lossy compressed images. Due to the lossy compression of JPEG format, the noise model in the Nam dataset and the PolyU dataset are more complex [21]. Therefore, fine-tuning the model on the new dataset in JPEG format is necessary. Benefitting from the fine-tuning, our proposed DRDN can also achieves much better performance than the methods mentioned in [30] as shown in Table 3.

### 4.4. Evaluation of the Threshold

We evaluate the results with different skip ratios via modifying the threshold in the gate module. We select multiple images for evaluation, and each image outputs multiple processed images with different skip ratios. As shown in Figure 5, the output images become sharper or smoother as the skip ratio change. For the images in Figure 5, we vote to choose the output image that best satisfies personal perception. The patches in the red boxes are the choices that are closest to the choices of the DRDN, and the patches in the green boxes are the voting choices. The patch in the yellow box is both the choice closest to the DRDN’s and the voting choice. Figure 5 shows that our researchers prefer to apply powerful denoising strength to denoise real-world noisy images. However, our researchers have also made different choices in choosing the image with the best perceptual quality. Some of our researchers think that images with little noise have better perception while others think that these images do not look good after zooming in. This is the reason why we want to provide adjustable denoising strength for real-world noise reduction.

Figure 6 shows the quantitative results of PSNR and SSIM with different skip ratios. It can be seen that the trend of PSNR and SSIM is obviously asymmetric. This may be because PSNR and SSIM are sensitive to noise, which makes over-smoothed images have better image quality than over-sharpened images. Furthermore, DRDN has a larger capacity when the skip ratio is lower, which makes it difficult for DRDN to over-smooth the images. In future research, we will try to expand the adjustable range of denoising strength.

### 4.5. Results of Reinforcement Learning

We also use reinforcement learning to fine-tune the model to further improve the skip ratio. We fine-tune the model trained on the RENOIR using reinforcement learning for 500 epochs. We visualize the skip ratio of different blocks of the model trained by supervised learning and reinforcement learning. Figure 7 shows the skip ratio of each block test on the RENOIR. It shows that the distribution of the skip ratio of the model trained by reinforcement learning is flatter than the model trained by supervised learning. Furthermore, the average skip ratio has also improved significantly.

We evaluate the model trained by supervised learning and reinforcement learning by PSRN, SSIM, and FLOPs, as shown in Table 4. Although the average FLOPs have been further reduced, the processing results have slightly deteriorated. This may be because the number of blocks in the DRDN is too small, which makes it not suitable for pruning the blocks by reinforcement learning. Moreover, we do not get an approximately continuously adjustable denoising strength via adjusting the threshold since the output probability of the gate module is relatively close to each other.

## 5. Conclusions

In this paper, we propose a DRDN model for noise reduction of real-world images. Our proposed DRDN makes full use of the properties of residual connection and deep supervision. We present a method to denoise images with different noise amounts and simultaneously reduce the average computational cost. The core idea of our method is to dynamically change the number of blocks involved in denoising to change the denoising strength via sequential decision. Moreover, our method can manually adjust the denoising strength of the model without fine-tuning the parameters.

## Figures and Tables

**Figure 1 sensors-19-03809-f001:**
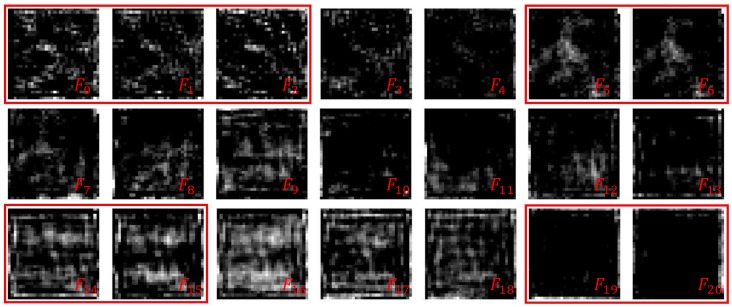
Feature map visualization of the RDBs. There are adjacent feature maps with higher similarity in the red squares.

**Figure 2 sensors-19-03809-f002:**
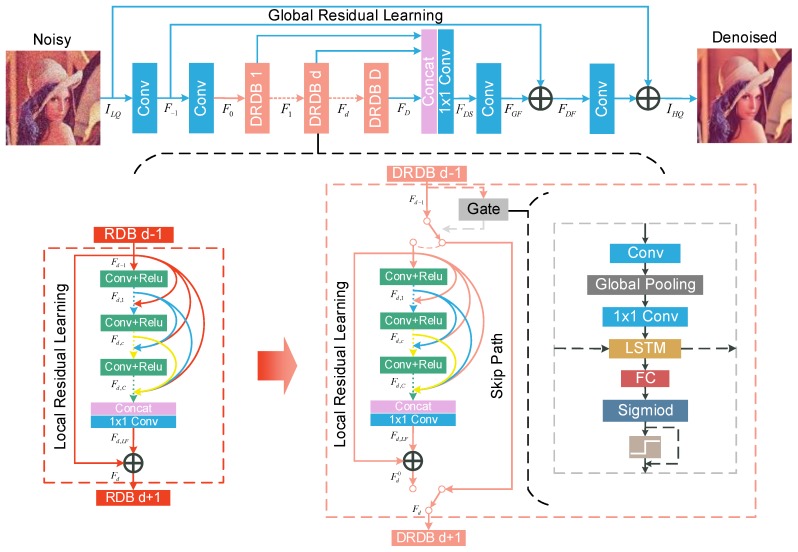
The top part presents the overall architecture of the DRDN. The bottom part shows the difference between the RDB and DRDB and how the gate module is inserted into the original block.

**Figure 3 sensors-19-03809-f003:**
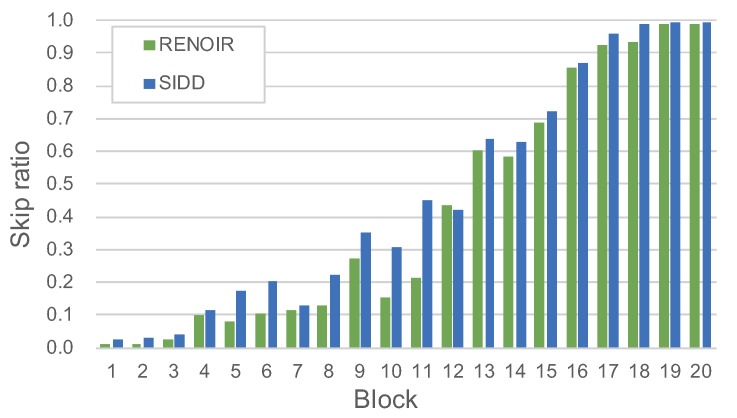
Skip ratio per block of the DRDN while testing on the RENOIR and the SIDD.

**Figure 4 sensors-19-03809-f004:**
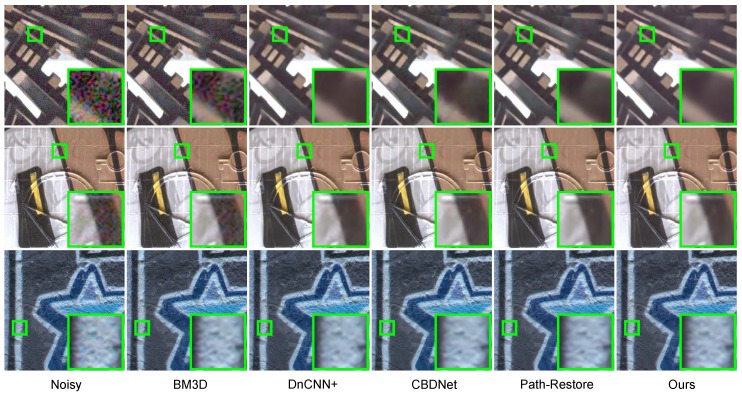
Qualitative results of real-world denoising on the Darmstadt Noise Dataset.

**Figure 5 sensors-19-03809-f005:**
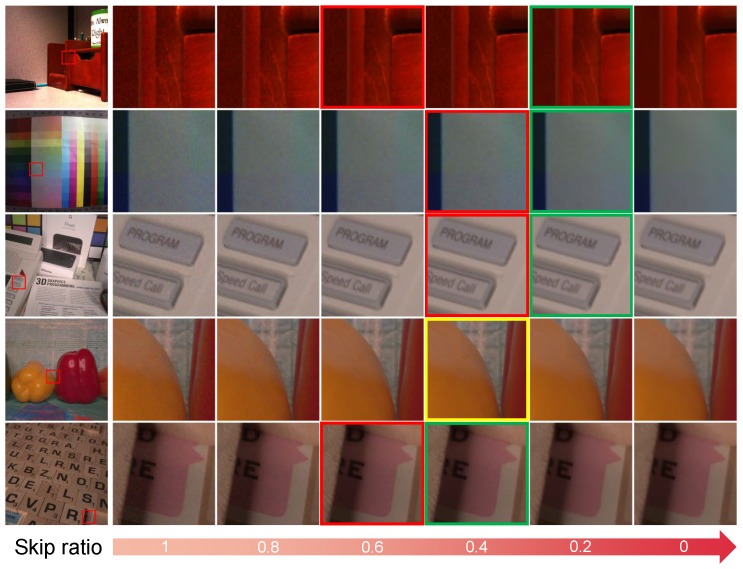
Denoising results vary as the skip ratio increases.

**Figure 6 sensors-19-03809-f006:**
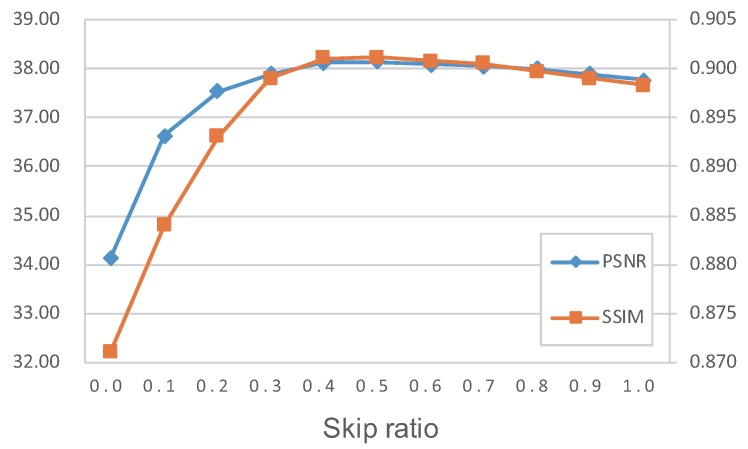
Trends of PSNR and SSIM with the skip ratio changing during testing on the RENOIR.

**Figure 7 sensors-19-03809-f007:**
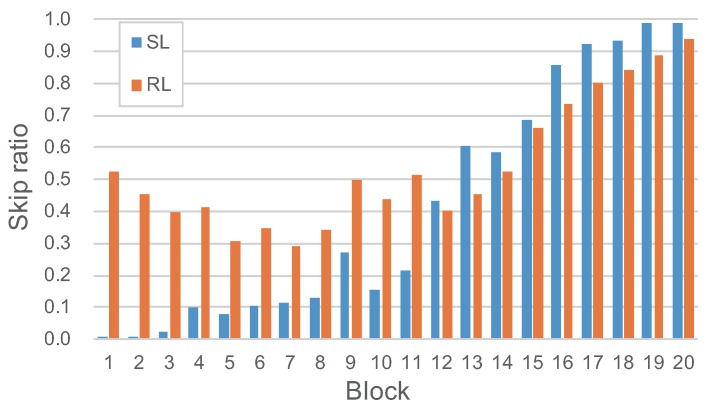
Skip ratio of the DRDN train via supervised learning or reinforcement learning on the RENOIR.

**Table 1 sensors-19-03809-t001:** Results of real-world denoising on the RENOIR [27] and the SIDD [28]. Best performance is in boldface.

Dataset	Method	PSNR	SSIM	Params (M)	FLOPs (G)	Latency (s)
RENOIR	RDN+	**38.17**	**0.9013**	**5.47**	105.5	0.63
	DRDN	38.12	0.9010	5.59	**61.18**	**0.49**
SIDD	RDN+	39.55	0.9399	**5.47**	105.5	0.63
	DRDN	**39.60**	**0.9401**	5.59	**53.00**	**0.42**

**Table 2 sensors-19-03809-t002:** Results of real-world denoising on the Darmstadt Noise Dataset [38]. Best performance is in boldface.

Method	PSNR	SSIM	Blind/Non-Blind
EPLL [13]	33.51	0.8244	Non-blind
TNRD [42]	33.65	0.8306	Non-blind
BM3D [12]	34.51	0.8507	Non-blind
MCWNNM [43]	37.38	0.9294	Non-blind
FFDNet+ [39]	37.61	0.9415	Non-blind
DnCNN+ [15]	37.90	0.9430	Blind
TWSC [44]	37.96	0.9416	Non-blind
CBDNet [21]	38.06	0.9421	Blind
PD [40]	38.40	0.9452	Blind
Path-Restore [41]	39.00	**0.9542**	Blind
DRDN	**39.40**	0.9524	Blind

**Table 3 sensors-19-03809-t003:** Results of real-world denoising on the Nam dataset [29] and the PolyU dataset [30]. Best performance is in boldface.

Dataset	Metric	EPLL [13]	BM3D [12]	TNRD [42]	DnCNN [15]	TWSC [44]	DRDN
Nam	PSNR	33.66	35.19	36.61	33.86	37.81	**38.45**
	SSIM	0.8591	0.8580	0.9463	0.8635	0.9586	**0.9626**
PolyU	PSNR	36.17	37.40	38.17	36.08	38.60	**38.96**
	SSIM	0.9216	0.9526	0.9640	0.9161	0.9685	**0.9691**

**Table 4 sensors-19-03809-t004:** Comparison of supervised learning and reinforcement learning on the RENOIR. Best performance is in boldface.

Method	PSNR	SSIM	FLOPs
DRDN+SL	**38.12**	**0.9010**	61.18
DRDB+RL	38.03	0.9003	**48.75**

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
