# Peer review of "Dynamic Residual Dense Network for Image Denoising"

_sensors, 2019, doi:10.3390/s19173809_

Round 1
Reviewer 1 Report
This paper presents a dynamic residual dense network to denoise. The main idea is to add gate modules to dynamically reduce the number of blocks in the denoising process. However, it is not well written in its methodology, and its results are incomplete and not promising. A major revision is suggested to improve this manuscript. The following give some critical points for the improvements: (1) Motivation explained in Section 1 should be improved. Arrangement and meaning of sub-figures in Figure 1 are not explained and therefore the high similarity of feature maps cannot be understood. Figure 2 shows nothing and can be removed. The sentence "This shows that replacing these RDBs with identity mappings" in Line 39 comes out without any reasonable explanation. (2) Literature survey in Section 2 is too weak. Many deep learning works for denoising shown in the Table 2, p. 8, are not discussed in Section 2. Besides, some recent works such as Fully Symmetric Convolutional Network for Effective Image Denoising should be reviewed in Section 2. (3) Methodology described in Section 3 is not well written. Lines 97-100 and Line 118 give some descriptions, such as feature extraction module and other names of blocks and modules, that do not appear in Figure 3. That is a great confusing. The training algorithm in Section 3.3 is not clear. The meaning of each stage is unclear. Gradient propagation in stage is not defined. A weird "Optional" paragraph comes out without any explanation, also without details of the reinforcement algorithm. (4) Experiments described in Section 4 are incomplete and give less details for the proposed DRDN method. (a) The hyperparameters in Section 4.1 are given without any explanation. The convergence of training curves for different training settings should be given to help the explanation. (b) RDN+ in line 188 is not defined. (c) Table 1 should be elaborated with more explanations, especially for the FLOPs. FLOPs should vary for each noisy images, especially for DRDN the variations should be great. It is not clear how to get the FLOPs in Table 1. Is it "average" FLOPs? And how the FLOPs varies corresponding to different noise levels. More statistics of FLOPs should be given and discussed. (d) Figure 4 should be elaborated with more explanations, especially for "skip ratio" and "deep supervision". The term "skip ration" is not defined and discussed. Lines 203-207 attribute the high skip ratio of latter block to deep supervision, which is not clear. More details and explanations are necessary. (e) Noise levels in Sections 4.2 and 4.3 are not described and discussed. There shall be many statistics of experimental results corresponding to different noise levels.
Reviewer 2 Report
The authors propose an improvement over Residual Dense Networks (RDB), by dynamically skipping some residual modules and in this way improving computational performance. The method also allows for the user to have some control over the intensity of the denoising operation.
The problem is relevant and the manuscript is well written and sounds technically correct. My main concerns are with the innovation of the proposal and the experimental analysis.
The change proposed in DRDB algorithm with respect to the original RDB method is incremental, based on the addition of a mechanism of LSTM and other classical strategies in neural networks. To counterbalance such absence of a more in-depth advance in theoretical terms, I suggest for the authors to include a more robust theoretical analysis, for example, explaining how the proposed dynamic module is capable of identifying the strength of the noise and how the gradient computation is affected by this addition.
The authors also show results on only one database (DND), which is widely used for benchmark purposes in denoising task, but they could carry out experiments on other databases, for example, BSD68 [1], BSDS [2], Set5 / Set14 [3], and others. Other recently published methods should be ideally compared in quantitative terms or, at least, mentioned in the text. Examples are the universal denoising network [4], deep boosting [5] or even others that are not so recent but are classically used in benchmarks, like EPLL [6].
Finally, although the authors show the method performance in terms of FLOPs, it would be interesting to list the runtime on a real hardware.
REFERENCES
[1] S. Roth and M. J. Black. Fields of experts. International Journal of Computer Vision, 82(2):205, 2009.
[2] D. Martin, C. Fowlkes, D. Tal, and J. Malik. A database of human segmented natural images and its application to evaluating segmentation algorithms and measuring ecological statistics. In Proc. IEEE Int. Conf. Computer Vision, pages 416–423, 2001.
[3] Chen, Y., Pock, T.: Trainable nonlinear reaction diffusion: a flexible framework for fast and effective image restoration. IEEE Trans. Patt. Anal. Mach. Intell. 39(6), 1256 (2017)
[4] Lefkimmiatis, S. Universal Denoising Networks : A Novel CNN Architecture for Image Denoising CVPR 2018. DOI: 10.1109/CVPR.2018.00338
[5] C. Chen et al. Real-world Image Denoising with Deep Boosting. IEEE Transactions on Pattern Analysis and Machine Intelligence ( Early Access ). DOI: 10.1109/TPAMI.2019.2921548
[6] D. Zoran and Y. Weiss. From learning models of natural image patches to whole image restoration. In Proc. IEEE Int. Conf. Computer Vision, pages 479–486. IEEE, 2011.
Round 2
Reviewer 1 Report
The authors revise the manuscript according to most of my comments. However, there is still no noise-level analysis of experiments and therefore minor revision to the manuscript is still necessary. Comments are given as follows:
(1) Line 8 in p.1: "the noise level" should be corrected to be "the noise amount".
(2) Line 62 in p.2: "the noise level" should be corrected to be "the noise amount".
(3) Line 259 in p. 8: "the noise level" should be corrected to be "the noise amount".
(4) Line 261 in p. 8: "different noise levels" should be corrected to be "different noise amounts".
(5) Line 338 in p. 12: "different noise levels" should be corrected to be "different noise amounts".
Author Response
We are very sorry for the deviation caused by the misuse of terminology. We have modified the manuscript based on your comments as follows:
Point 1: Line 8 in p.1: "the noise level" should be corrected to be "the noise amount". 

Response 1: We have corrected "the noise level" to "the noise amount" in line 8 (page 1).
Point 2: Line 62 in p.2: "the noise level" should be corrected to be "the noise amount".
Response 2: We have corrected "the noise level" to "the noise amount" in line 62 (page 2).
Point 3: Line 259 in p. 8: "the noise level" should be corrected to be "the noise amount".
Response 3: We have corrected "the noise level" to "the noise amount" in line 259 (page 8).
Point 4: Line 261 in p. 8: "different noise levels" should be corrected to be "different noise amounts".
Response 4: We have corrected "different noise levels" to "different noise amounts" in line 261 (page 8).
Point 5: Line 338 in p. 12: "different noise levels" should be corrected to be "different noise amounts".
Response 5: We have corrected "different noise levels" to "different noise amounts" in line 338 (page 12).
Reviewer 2 Report
The authors appropriately addressed my main concerns and the manuscript can be accepted in its present form.
Author Response
Thank you very much for your great efforts on our manuscript and all your comments are helpful to us.